# Reverse Genetics and Artificial Replication Systems of Borna Disease Virus 1

**DOI:** 10.3390/v14102236

**Published:** 2022-10-12

**Authors:** Takehiro Kanda, Keizo Tomonaga

**Affiliations:** 1Laboratory of RNA Viruses, Department of Virus Research, Institute for Life and Medical Sciences, Kyoto University, Kyoto 606-8507, Japan; 2Department of Molecular Virology, Graduate School of Medicine, Kyoto University, Kyoto 606-8507, Japan; 3Laboratory of RNA Viruses, Department of Mammalian Regulatory Network, Graduate School of Biostudies, Kyoto University, Kyoto 606-8507, Japan

**Keywords:** Borna disease virus, reverse genetics, orthobornaviruses, viral replication and transcription

## Abstract

Borna disease virus 1 (BoDV-1) is a neurotropic RNA virus belonging to the family *Bornaviridae* within the order *Mononegavirales*. Whereas BoDV-1 causes neurological and behavioral disorders, called Borna disease (BD), in a wide range of mammals, its virulence in humans has been debated for several decades. However, a series of case reports in recent years have established the nature of BoDV-1 as a zoonotic pathogen that causes fatal encephalitis in humans. Although many virological properties of BoDV-1 have been revealed to date, the mechanism by which it causes fatal encephalitis in humans remains unclear. In addition, there are no effective vaccines or antiviral drugs that can be used in clinical practice. A reverse genetics approach to generating replication-competent recombinant viruses from full-length cDNA clones is a powerful tool that can be used to not only understand viral properties but also to develop vaccines and antiviral drugs. The rescue of recombinant BoDV-1 (rBoDV-1) was first reported in 2005. However, due to the slow nature of the replication of this virus, the rescue of high-titer rBoDV-1 required several months, limiting the use of this system. This review summarizes the history of the reverse genetics and artificial replication systems for orthobornaviruses and explores the recent progress in efforts to rescue rBoDV-1.

## 1. Introduction

Borna disease virus 1 (BoDV-1) is a neurotropic virus that causes neurological and behavioral disorders, called Borna disease (BD), in a wide range of mammals, including horses, sheep and other mammals [1,2,3]. Together with Borna disease virus 2 (BoDV-2), which was isolated from a pony showing severe and incurable neurological symptoms [4,5], BoDV-1 has been classified as the species *Orthobornavirus bornaense* [6]. Whether or not BoDVs are zoonotic pathogens in humans has been debated for the past few decades, but several recent studies have reported a relationship between BoDV-1 infection and fatal encephalitis in humans. In 2018, BoDV-1 RNA and antigens were detected in a previously healthy 25-year-old man who died of encephalitis [7]. In addition, three solid-organ transplant recipients who received organs from a donor living in a BoDV-1-endemic region showed neurological symptoms [8]. Two patients died of encephalitis, and BoDV-1 RNA and antigens were detected in both patients [8]. Moreover, retrospective studies reported that BoDV-1 RNA and antigens were detected in 14 brain tissues taken from patients who died of an unclassified encephalitis [9,10,11]. To date, nearly 40 cases of human fatal encephalitis caused by BoDV-1 infection have been reported in the region of central Europe, an endemic region for BoDV-1 [12,13,14]. In addition, a new species of mammalian bornavirus, variegated squirrel bornavirus 1 (VSBV-1), which has been classified as the species *Orthobornavirus sciuri*, was discovered when variegated squirrel breeders died of encephalitis in 2015 [15], and a total of six cases of human VSBV-1 infection have been reported to date [16,17,18]. Based on this recent evidence, mammalian bornaviruses, particularly BoDV-1 and VSBV-1, must be considered as potentially lethal zoonotic pathogens in humans [19]. In addition, given its close phylogenetic relationship with BoDV-1, BoDV-2 possesses the potential to cause fatal encephalitis in humans, indicating that further investigations are required in order to understand the epidemiology, biology and pathogenicity of mammalian bornaviruses and to develop diagnostic tools and effective vaccines and antiviral drugs.

A reverse genetics approach to generating replication-competent recombinant viruses from full-length cDNA clones is a powerful method that can be used to not only understand viral properties, such as the function of viral genes, the viral replication cycle and viral pathogenicity, but also to develop vaccines and antiviral drugs. Furthermore, reverse genetics technologies are widely applied to viral vector systems that can transduce foreign genes into specific cells or organs. The rescue of replication-competent recombinant BoDV-1 (rBoDV-1) was first reported in 2005 [20]. In this system, however, rBoDV-1-infected cells were not detected for two weeks, and subsequent long-term cell culture was required to spread the infection and rescue high-titer rBoDV-1. In addition, this inefficient system posed difficulties for the rescue of rBoDV-1 harboring arbitrary mutations that might decrease the fitness of viral survival, limiting the availability of this system [21]. Thus, considerable improvements were required in order to increase the utility of reverse genetics systems for BoDV-1.

Here, we compile the history of the reverse genetics systems for BoDV-1 and describe the recent progress. After two decades of laborious efforts on the part of bornavirus researchers, an efficient reverse genetics system, enabling the rescue of rBoDV-1 within the space of a few days, has been developed.

## 2. Orthobornavirus

Orthobornavirus is a non-segmented, negative-strand RNA virus in the family *Bornaviridae* within the order *Mononegavirales*, which includes the prototype virus of this family, BoDV-1 [6]. The genomic RNA of orthobornavirus encodes at least six viral proteins: the nucleoprotein (N), accessary protein (X), phosphoprotein (P), matrix protein (M), glycoprotein (G) and large protein (L) in the 3′ to 5′ orientation of the genome [22,23,24]. Based on studies of BoDV-1, many properties of these viral proteins have been elucidated. N encapsulates viral genomic and antigenomic RNAs forming the N-RNA complex, nucleocapsid. L, which encodes an RNA-dependent RNA polymerase, interacts with its cofactor P, forming a viral polymerase complex. The minimum viral replication and transcription unit, the viral ribonucleoprotein (vRNP), is constructed through interactions between the nucleocapsid and polymerase complex [25,26,27]. The other two structural proteins, the matrix protein (M) and glycoprotein (G), are associated with viral particle formation [28]. M plays critical roles in viral assembly and structural formation [29,30], and G serves pivotal functions in the processes of receptor recognition, cell entry and membrane fusion [31,32,33]. While M is involved in vRNP through interactions with P, it does not affect the efficiency of viral replication or transcription [34]. The only nonstructural protein X is known to associate with vRNPs in infected cells [35,36], but its roles in the BoDV-1 replication cycle remain unclear.

## 3. Principle of the Artificial Replication System of Bornaviruses

### 3.1. Artificial Viral Replication of Mononegaviruses

Viruses with a genome that consists of non-segmented, negative-strand RNA are classified as belonging to the order *Mononegavirales*. Although certain viruses carry a few additional virus-specific genes, such as the X gene in BoDV-1, mononegaviruses commonly encode N, P, M, G and L in the 3′ to 5′ orientation in the genome. The reverse genetics technology used for many mononegaviruses was established in the late 1990s [37]. Prior to recovering recombinant viruses from cDNA clones, a minireplicon system was constructed to identify the viral proteins essential for the replication and transcription of viral RNA. Experiments using minireplicons of representative mononegaviruses, such as vesicular stomatitis virus (VSV) [38], Sendai virus (SeV) [39] and human respiratory syncytial virus (hRSV) [40], revealed that N, P and L were sufficient for viral replication and transcription, and they were found to be the minimal components of vRNP. Through these findings, the rescue of recombinant rabies virus (RABV) using a plasmid encoding the full-length RABV antigenome and three helper plasmids expressing N, P and L was achieved and reported in 1994, as was the first case among the mononegaviruses [41].

### 3.2. Minireplicon System of Orthobornaviruses

The first minireplicon system of BoDV-1 was reported by Perez et al. in 2003 [25] (Figure 1). In this system, a murine RNA polymerase I (mPol I) promoter-driven minireplicon plasmid expressing a chloramphenicol acetyltransferase (CAT) reporter gene flanked by the 3′ and 5′ untranslated regions of BoDV-1 in the negative-sense orientation was co-transfected with three helper plasmids expressing N, P and L into BHK-21 cells. The expression of CAT was detected only when three helper plasmids expressing N, P and L were transfected simultaneously, indicating that they were necessary and sufficient for the replication and transcription of BoDV-1, as well as other mononegaviruses. Shortly thereafter, Schneider et al. reported a BoDV-1 minireplicon system, in which a T7 RNA promoter-driven BoDV-1 minireplicon plasmid and BSR-T7 cells stably expressing T7 RNA polymerase were used [26] (Figure 1c). Interestingly, although these studies used different RNA polymerases to generate BoDV-1 minireplicons, the expression level of CAT was maximized when the amount of transfected helper plasmid encoding N was approximately 10- to 20-fold greater than that encoding P [25,26]. On the other hand, the ratio of helper plasmid encoding L to that encoding P did not affect the reporter activity, indicating that the N-to-P stoichiometry regulates BoDV-1 polymerase activity [26]. To further elevate the synthesis of the BoDV-1 minireplicon in the nucleus, where BoDV-1 replicates and is transcribed, Yanai et al. developed an RNA polymerase II (Pol II) promoter-driven minireplicon system [42]. In addition to the use of the CAG promoter, a combination of a cytomegalovirus early enhancer with a chicken β-actin promoter, and the SV40 nuclear import sequence was inserted into the minireplicon plasmid to facilitate plasmid nuclear import (Figure 1c). Although the human Pol I promoter-driven minireplicon was only applicable to human cell lines, the novel minireplicon was functional in several mammalian cell lines and successfully elevated the expression of the BoDV-1 minireplicons, expanding the practical value of the BoDV-1 minireplicon system.

The minireplicon system is applicable to the evaluation of the efficacy of antiviral drugs targeting viral RNA synthesis. Tokunaga et al. demonstrated that a nucleoside analog, favipiravir (T-705), efficiently inhibited the viral polymerase activity of BoDV-1 using a minireplicon system [43]. Reuter et al. also demonstrated that another nucleoside analog, ribavirin, inhibited the viral RNA synthesis of an avian bornavirus, parrot bornavirus 4 (PaBV-4), using PaBV-4 N, P and L together with the BoDV-1 minireplicon in chicken DF-1 cells [44].

Recently, Komorizono et al. established a PaBV-4 minireplicon system using PaBV-4 N, P and L together with the PaBV-4 minireplicon [45]. Interestingly, while the BoDV-1 minireplicon system was functional in both mammalian cells and avian cells, the PaBV-4 minireplicon system was functional only in avian cells. A careful comparison of each component revealed that the nuclear localization signal (NLS) in PaBV-4 N was only weakly functional in mammalian cells, which disrupted the nuclear import of PaBV-4 N and inhibited the polymerase activity of PaBV-4 in mammalian cells.

### 3.3. Reverse Genetics Technology of BoDV-1

The rescue of replication-competent rBoDV-1 was first reported by Schneider et al. in 2005 [20] (Figure 2). In this study, the BoDV-1 antigenome plasmid, expressing the full-length BoDV-1 antigenome flanked by a hammerhead ribozyme and a hepatitis delta virus ribozyme under the control of the T7 promoter, was co-transfected with three helper plasmids expressing N, P and L into BSR-T7 cells. Three days after transfection, the cells were cocultured with G418-resistant Vero cells and passaged under continuous G418 selection. As a result, the rBoDV-1 infection was first detected in cocultured the Vero cells approximately 15–20 days after plasmid transfection and steadily spread thereafter. Although replication-competent rBoDV-1 was successfully rescued using this system, long-term cell culture was required in order to spread the infection and obtain high-titer rBoDV-1. Thus, further investigations were necessary in order to shorten the period of time required to obtain rBoDV-1.

To improve the system, Martin et al. replaced the T7 promoter encoded in the BoDV-1 antigenome plasmid with the Pol II promoter [46] (Figure 2b). The use of the Pol II promoter obviated the necessity of the exogenous expression of T7 RNA polymerase and enabled the use of highly transfectable HEK293T cells for the plasmid transfection. Therefore, the use of the Pol II promoter and HEK293T cells increased the synthesis of the full-length BoDV-1 antigenome from the plasmid, resulting in a 20-fold increase in the rescue efficiency compared to that of the T7 promoter.

Shortly thereafter, Ackermann et al. reported three amino acid mutations necessary for BoDV-1 to acquire infectivity in mice [47] (Figure 2b). In this study, the authors isolated BoDV-1, which propagated efficiently and induced neurological diseases in mice after four consecutive passages in the mouse brain. A sequence analysis of the mouse-adapted BoDV-1 revealed that three nonsynonymous mutations had accumulated: one in P, R66K, and two in L, L1116R and N1398D. rBoDV-1, harboring these mutations, demonstrated a significantly accelerated growth ability, shortening the cell culture period required to spread the infection into Vero cells.

We also reported a unique approach to increasing the rescue efficiency of rBoDV-1. Because heat stress was reported to enhance the rescue efficiency of recombinant measles virus (MeV) [48], a representative mononegavirus, we examined whether the rescue efficiency of rBoDV-1 is also increased by heat stress [49]. The study showed that heat stress enhanced the viral replication and transcription not only in persistently BoDV-1-infected cells but also in HEK293T cells that was transfected with BoDV-1 antigenome plasmid and helper plasmids, indicating that heat stress is an effective approach to increasing the rescue efficiency of rBoDV-1.

One of the biggest advantages of reverse genetics technology is its ability to generate recombinant viruses harboring arbitrary gene insertions and deletions. To exploit this advantage for virological research on BoDV-1, rBoDV-1 expressing a fluorescent reporter gene was generated using reverse genetics technology. Schneider et al. and Daito et al. generated rBoDV-1 expressing GFP by inserting a GFP expression cassette into the 5′ end of the genome (rBoDV-1 5′-GFP) [50] and between the P and M genes (rBoDV-1 P/M-GFP) [51], respectively (Figure 3). Although both rBoDV-1 5′-GFP and rBoDV-1 P/M-GFP showed a similar growth ability, which was slightly slower than that of rBoDV-1 without an additional expression cassette, rBoDV-1 P/M-GFP expressed higher levels of GFP than rBoDV-1 5′-GFP. In addition, rBoDV-1s expressing DsRed and luciferase were generated using an expression cassette inserted between the P and G genes, which were designated as rBoDV-1 P/M-DsRed and rBoDV-1 P/M-Luc, respectively, without the attenuation of the rescue efficiency or growth ability [51]. The use of rBoDV-1 expressing a reporter gene makes it easy to trace viral propagation both in vitro and in vivo [52]. In addition, by inserting therapeutic genes into the expression cassette, rBoDV-1 was developed as a novel and unique viral vector [53] (described in Section 5).

To cite another example, Charlier et al. generated rBoDV-1 expressing a Flag tag and a tetracysteine tag (TCT)-fused P (rBoDV-1 P-Flag-TCT) [54]. The labeling of the TCT with a fluorescent biarsenical derivative, FlAsH, allowed for the analysis of the real-time intracellular distribution and trafficking of P. Furthermore, the use of rBoDV-1 P-Flag-TCT enabled the visualization of the cell-to-cell BoDV-1 transmission and transportation of incoming viral protein towards the nucleus, followed by the construction of viral inclusions in the nucleus.

## 4. Latest Progress in Reverse Genetics System of BoDV-1

### 4.1. Replacement of the Helper Plasmid of BoDV-1 N with That of BoDV-2 N

Although we and other groups have addressed several challenges in order to improve the reverse genetics system of BoDV-1, the rescue efficiency has remained low, and several weeks are necessary to obtain high-titer rBoDV-1 [55]. Thus, to further improve the system, we focused on BoDV-2, another genotype of *Orthobornavirus bornaense*. In a previous report, BoDV-2 has been shown to rapidly propagate in cultured Vero cells without causing obvious cytopathogenic effects [4], suggesting that BoDV-2 may provide clues that can be used to improve the reverse genetics system of BoDV-1. To examine this possibility, we constructed expression plasmids for BoDV-2 N, P and L by referring to the nucleotide sequence of BoDV-2 strain No/98, as registered in NCBI GenBank [5]. Although the expression of BoDV-2 N and P was confirmed by Western blotting, that of BoDV-2 L was not [55]. Previously, it was reported that the nucleotide sequence of BoDV-2 L might not be correct [5]. Thus, we examined the effects of BoDV-2 N and P on the polymerase activity of BoDV-1 L. In the minireplicon assay, the replacement of the helper plasmid of BoDV-1 N with that of BoDV-2 N significantly increased the expression level of the reporter gene encoded in the BoDV-1 minireplicon [55]. Through reverse genetics, the replication and transcription of BoDV-1 were enhanced by using BoDV-2 N as a helper plasmid. In addition, a chimeric rBoDV-1, which possessed BoDV-2 N instead of BoDV-1 N, produced high levels of viral mRNA and genomic RNA compared to the parental rBoDV-1. These findings indicated that BoDV-2 N has the potential to upregulate the RNA synthesis activity of BoDV-1. As a result, by using BoDV-2 N as a helper plasmid, rBoDV-1 was rescued directly from transfected HEK293T cells due to increased viral replication and transcription. Moreover, the period required to spread the rBoDV-1 infection to cocultured Vero cells was shortened by 3 weeks, resulting in the rescue of high-titer rBoDV-1 within the space of a few weeks [55]. 

### 4.2. Exogeneous Expression of M and G

Next, to further increase the rescue efficiency of rBoDV-1, we focused on M and G, viral proteins essential for infectious particle formation. Because the M and G genes of BoDV-1 are transcribed together with the L gene, as a polycistronic transcript [24,56], and the translation of G depends on leaky scanning and ribosomal re-initiation mechanisms [57], it was predicted that the expression levels of M and G would be restricted in BoDV-1-infected cells. Thus, we examined whether the exogenous expression of M and G affected the infectious particle production of BoDV-1. When a moderate amount of both M and G, not either protein alone, was expressed exogenously in cells persistently infected with BoDV-1, the rescued viral titer was significantly increased. In addition, the spread of the BoDV-1 infection was accelerated in the Vero-MG cells stably expressing both M and G, compared to the parental Vero cells. Moreover, the use of M and G expression plasmids together with helper plasmids for N, P and L in the reverse genetics markedly increased the rescue efficiency of rBoDV-1, enabling a certain amount of rBoDV-1 to be obtained directly from the transfected HEK293T cells within 3 days of transfection [58]. In these experiments, while the moderate exogenous expression of both M and G facilitated the infectious particle production, the copy numbers of viral genomic RNA and viral mRNA were not affected, indicating that the expression levels of M and G are not sufficient for efficient viral particle formation in BoDV-1-infected cells. On the other hand, the overexpression of either M or G resulted in a significant reduction in the viral titer. Because the overexpression of M decreased the cell viability, and the overexpression of G resulted in the accumulation of immature G on the virions [30,58], an appropriate level of the exogenous expression of both M and G is required in order to maximize the viral particle production of BoDV-1. Thus, the use of novel Vero-MG cells, in which the expression levels of M and G can be controlled, is expected to accelerate viral propagation and facilitate viral particle production, leading to the rescue of high-titer rBoDV-1.

## 5. Application of the Artificial Replication System of BoDV-1

In contrast to other RNA viruses, BoDV-1 replicates and is transcribed in the nucleus and establishes a persistent infection without causing obvious cytopathic effects [22,59]. To achieve this, BoDV-1 constructs its vRNPs using host chromatin as a scaffold. Due to the stable interactions of vRNPs on the chromosomes during the cell cycle, vRNPs are separated into each daughter cell with the chromosomes, enabling the establishment of a persistent infection, even in the dividing cells without integration into the host genome [60,61,62].

These unique features of BoDV-1 were exploited in order to develop a novel viral vector that achieves long-term gene expression without causing cytotoxicity or genome toxicity. Using reverse genetics technology, rBoDV-1 with a foreign gene expression cassette between the P and M genes was generated and designated as an RNA-virus-based episomal vector (REVec) [51] (Figure 3). To alleviate the potential pathogenicity of the replication-competent REVec, transmission-deficient REVec was generated by deleting the G gene from the genome (ΔG-REVec) [51] (Figure 3). Furthermore, REVec that cannot form virus-like particles was generated by deleting both the M and G genes (ΔMG-REVec) [63] (Figure 3). Transduction with either ΔG-REVec or ΔMG-REVec achieved long-term gene expression in transduced cells without spreading the infection to adjacent cells, demonstrating that they were ideal platforms for the safe use of REVec.

**Figure 3 viruses-14-02236-f003:**
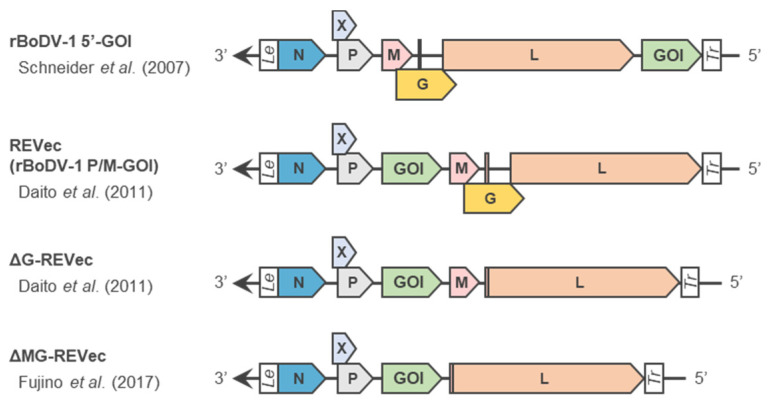
Engineering of rBoDV-1. Schematic representation of BoDV-1 genomes that harbor an additional gene expression cassette and gene deletion. rBoDV-1 5′-GOI harbors an additional gene expression cassette at the 5′ end of the genome [50]. REVec (rBoDV-1 P/M-GOI) harbors an additional gene expression cassette between the P and M genes [51]. ΔG-REVec [51] and ΔMG-REVec [63] lack the G gene and both the M and G genes, respectively. An inserted gene of interest (GOI) is colored green.

Compared to the commonly used virus vectors, one of the biggest advantages of REVec is its highly efficient transduction into stem cells, such as human mesenchymal stem cells (MSCs) and induced pluripotent stem cells (iPSCs) [64,65]. Ikeda et al. demonstrated that the transduction of MSCs and iPSCs with REVec achieved long-term transgene expression, maintaining the adipogenicity of MSCs and the pluripotency of iPSCs [64]. Komatsu et al. demonstrated that REVec is a potential tool that can be used to induce the differentiation of iPSCs in skeletal muscle cells [65]. Although further engineering is required, this evidence indicates that the REVec system is a promising application in the fields of regenerative medicine and gene and cellular therapy.

## 6. Perspectives

### 6.1. Elucidation of the Replication Mechanisms of Orthobornaviruses

Despite the existence of previous studies, it is difficult to determine exactly how each viral protein contributes to the replication of orthobornaviruses and, in particular, precisely what an accessary protein X contributes to it. BoDV-1 X is the only viral protein that is not involved in the viral particle [36], but it participates in the vRNPs through interactions with P [35]. To date, we and other groups have reported that X strongly suppresses the polymerase activity of BoDV-1 by directly interacting with P in a minireplicon system [25,26,42,66]. Although X does not have a canonical nuclear localization signal (NLS), it is imported into the nucleus through interactions with host importin α [67]. We also reported that X interacts with a host chaperone protein, 71-kDa heat shock cognate protein (Hsc70), and is translocated into the nucleus in association with the nuclear accumulation of Hsc70 under heat shock stress conditions [68]. In the nucleus, P binds to X in competition with Hsc70, displacing Hsc70 from X. The binding of X to P activates a nuclear export signal (NES) encoded in P, resulting in interactions with a host exportin, chromosome region maintenance protein 1 (CRM1), and the exportation of X and P to the cytoplasm [69,70]. In addition, these functions of X, the negative regulation of viral RNA synthesis and the nuclear exportation of P, are conserved in mammalian and avian bornaviruses [71]. These findings imply that the inhibitory effect of X might be exerted by sequestering P to the cytoplasm. However, while a P mutant lacking a functional NES could fully support viral replication and transcription in the minireplicon system, X exerted a comparable inhibitory effect on the polymerase activity in cells transfected with either the wild-type P or mutant P [70], indicating that the significance of the nuclear exportation of P by X to the regulation of polymerase activity is uncertain. In addition, whether X plays other roles in the BoDV-1 infection cycle remains unclear. Recently, some groups reported that X interacts with a mitochondrial antiviral-signaling protein (MAVS) in the mitochondria and inhibits the induction of the apoptosis of BoDV-1-infected cells [72,73,74]. Szelechowski et al. reported that X possesses strong axoprotective properties, protecting the neurons from degeneration [75]. These findings indicate that X might play a critical role in exerting pathogenic effects on BoDV-1-infected animals, including humans. Thus, it is worth exploring the function of X in the BoDV-1 replication cycle from the perspective of a strategy that can be used to establish persistent infection, as well as the mechanism involved in the viral pathogenicity. 

Elucidating the roles of X in the BoDV-1 replication cycle has not been easy due to the lack of a technique for recovering X-deficient rBoDV-1. Although Poenisch et al. attempted to rescue X-deficient rBoDV-1 by inactivating the initiation codon for X, X-deficient rBoDV-1 was not obtained [76]. On the other hand, while they rescued X-deficient rBoDV-1 by inserting an artificial X gene expression cassette into the 5′ end of the viral genome, both the expression level of X and the viral growth ability were severely attenuated [76], indicating that the tightly controlled expression of X is mandatory for the BoDV-1 replication cycle. We believe that the developed reverse genetics system can contribute to the rescue of X-deficient rBoDV-1, leading to a deeper understanding of the molecular biology of BoDV-1. Recently, we developed a system that effectively controls the expression level of the gene of interest (GOI) inserted into the rBoDV-1 genome by using the cis-acting, self-cleaving riboswitch L2bulge9 (L2b9) [77]. We believe that, if such a system can be used to generate rBoDV-1 with a tightly controlled X expression, it will not only answer the question regarding the function of the X gene but will also help us to further understand the mechanisms of orthobornavirus replication.

### 6.2. Reverse Genetics Systems of Other Mammalian and Avian Bornaviruses

As a result of several improvements, a reverse genetics system for BoDV-1, through which rBoDV-1 can be rescued within a few days of transfection, was established (Figure 4). However, as described in the introduction (Section 1), it was discovered that both BoDV-1 and VSBV-1 are lethal zoonotic pathogens in humans. In addition, BoDV-2 likely has pathogenic potential in humans, suggesting that research should be performed equally for all mammalian bornaviruses. To this end, a reverse genetics system for BoDV-2 and VSBV-1 should be established as soon as possible. Interestingly, we demonstrated that while BoDV-2 P alone did not affect the polymerase activity of BoDV-1 in a minireplicon assay, the polymerase activity and growth ability of a chimeric rBoDV-1 harboring BoDV-2 X and P instead of those of BoDV-1 were markedly attenuated, indicating the distinct roles of BoDV-2 X and P as compared to those of BoDV-1 in the viral replication cycle [55]. A comparison of each viral gene among the mammalian bornaviruses using reverse genetics technology could be a powerful approach for identifying the mechanisms of replication, persistent infection and pathogenicity of mammalian bornaviruses.

In addition, after the discovery of the first avian bornavirus in 2008 [78,79], a variety of bornaviruses have been detected in birds [80]. In particular, PaBV-4, which is the most dominant genotype causing neurological disorders and proventricular dilatation disease (PDD), has been detected in a broad range of captive psittacines worldwide [81,82], resulting in great economic damage to the bird breeding industry. However, the mechanisms of transmission and pathogenicity remain unclear, and no effective antiviral drugs are available to date. Thus, to construct fundamental research bases, the establishment of a reverse genetics system for avian bornaviruses is also required. Although the genetic background of PaBV-4 is similar to that of BoDV-1, Komorizono et al. showed that PaBV-4 lacks replication capacity in mammalian cells, indicating that culture cells other than HEK293T cells and Vero cells are necessary in order to propagate recombinant avian bornaviruses. Fortunately, Komorizono et al. demonstrated that PaBV-4 and other avian bornaviruses can replicate in chicken-fibroblast-derived DF-1 cells and quail-fibrosarcoma-derived QT6 cells [45]. We believe that a reverse genetics system for avian bornaviruses can be established by imitating that of BoDV-1 using avian cells.

## Figures and Tables

**Figure 1 viruses-14-02236-f001:**
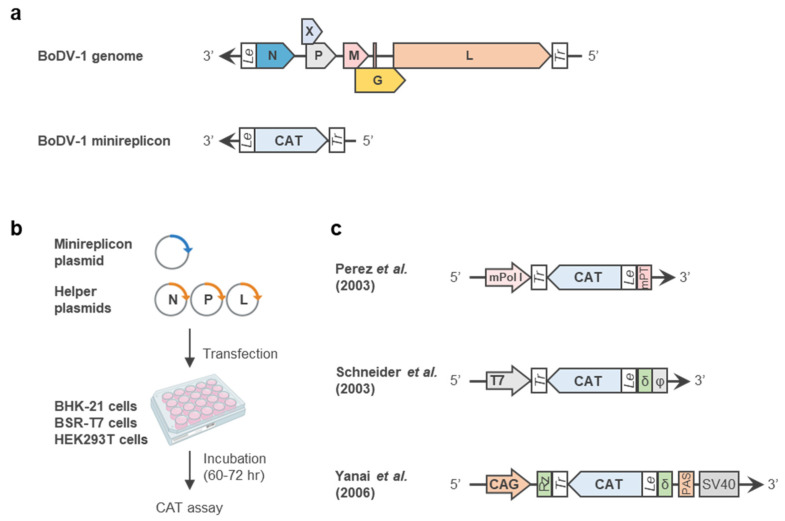
Development of BoDV-1 minireplicon technology. (**a**) Components of the BoDV-1 minireplicon. A reporter CAT gene is flanked by the 3′ and 5′ untranslated regions of BoDV-1, the leader sequence (Le) and trailer sequence (Tr), respectively, in the negative-sense orientation. (**b**) A protocol for the BoDV-1 minireplicon assay. A minireplicon plasmid encoding the BoDV-1 minireplicon and three helper plasmids expressing N, P and L were co-transfected into cultured cells. The level of CAT was titrated to measure the polymerase activity of BoDV-1 for 60 to 72 h post-transfection. (**c**) Configuration of BoDV-1 minireplicon plasmids. The minireplicon plasmid constructed by Perez et al. encodes the BoDV-1 minireplicon between the murine Pol I promoter (mPol I) and the murine Pol I terminator (mPT) [25]. The minireplicon plasmid constructed by Schneider et al. encodes the BoDV-1 minireplicon and a hepatitis delta virus ribozyme (δ) between the T7 promoter (T7) and the T7 terminator (φ) [26]. The minireplicon plasmid constructed by Yanai et al. encodes the BoDV-1 minireplicon flanked by a hammerhead ribozyme (Rz) and a hepatitis delta virus ribozyme (δ) between the CAG promoter and polyadenylation signal (PAS). The SV40 nuclear import sequence is inserted downstream of the PAS [42].

**Figure 2 viruses-14-02236-f002:**
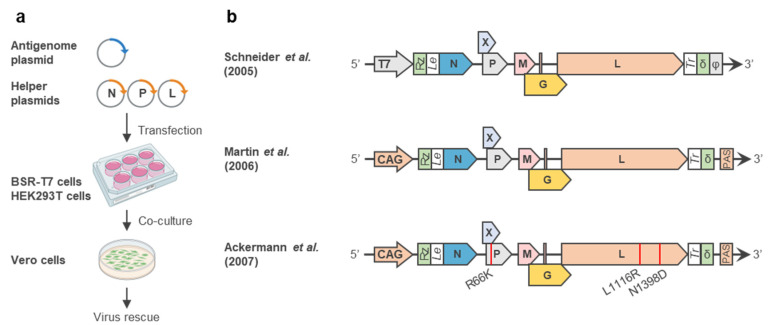
Development of a reverse genetics system for BoDV-1. (**a**) A protocol for BoDV-1 reverse genetics. A BoDV-1 antigenome plasmid encoding the full-length BoDV-1 antigenome and three helper plasmids expressing N, P and L were co-transfected into cultured cells. At 72 h post-transfection, the transfected cells were cocultured with Vero cells to spread the infection efficiently. The cells were passaged every 3 or 4 days until the rBoDV-1 infection spread to most Vero cells. (**b**) Configuration of the BoDV-1 antigenome plasmid. The antigenome plasmid constructed by Schneider et al. encodes a BoDV-1 full-length antigenome flanked by a hammerhead ribozyme (Rz) and a hepatitis delta virus ribozyme (δ) between the T7 promoter (T7) and the T7 terminator (φ) [20]. The antigenome plasmid constructed by Martin et al. encodes a BoDV-1 full-length antigenome flanked by a hammerhead ribozyme (Rz) and a hepatitis delta virus ribozyme (δ) between the CAG promoter and polyadenylation signal (PSA) [46]. The antigenome plasmid constructed by Ackermann et al. was developed from the plasmid constructed by Martin et al. by inserting a nonsynonymous mutation, R66K, and two nonsynonymous mutations, L1116R and N1398D, into the P and L genes of the BoDV-1 antigenome [47].

**Figure 4 viruses-14-02236-f004:**
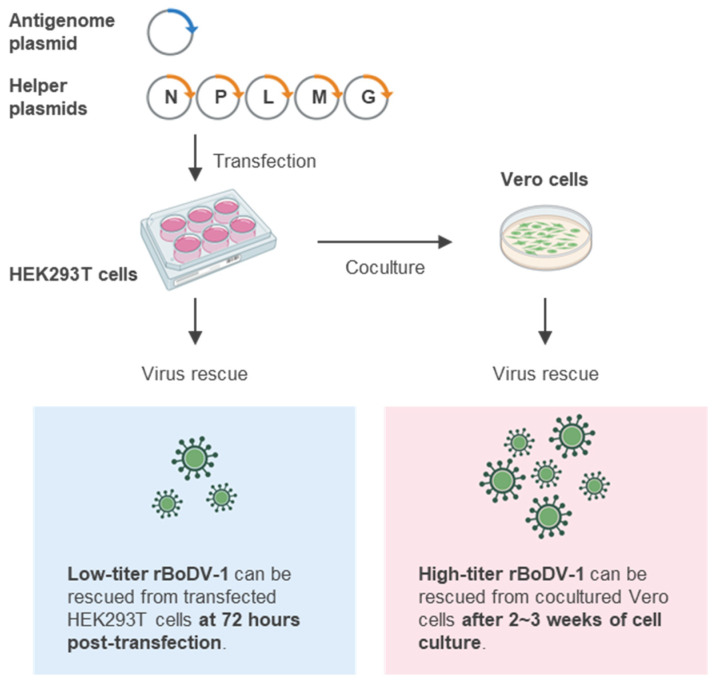
Improved method of BoDV-1 reverse genetics. A BoDV-1 antigenome plasmid and five helper plasmids were transfected into HEK293T cells. To rescue rBoDV-1 as quickly as possible, rescue rBoDV-1 directly from the transfected cells at 72 h post-transfection. To rescue high-titer rBoDV-1, coculture transfected HEK293T cells with Vero cells for a few weeks in order to spread the infection and rescue rBoDV-1 from cocultured Vero cells.

## Data Availability

Not applicable.

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
