# Peer review of "Reverse Genetics and Artificial Replication Systems of Borna Disease Virus 1"

_viruses, 2022, doi:10.3390/v14102236_

Round 1

Reviewer 1 Report

In this manuscript the authors present a review of reverse genetics of Avian Borne virus I. A similar review was published by the same authors in Current Opinion in Virology (Vol 44, pages 42-48) in October 2020. Compared to earlier review, this report appears to be more comprehensive. But overall, this manuscript does not add much over what was reviewed a few years earlier.

Minor suggestions:

Line 38-39 Add reference

Line 58: what do you mean "crucial to regenerative medicine"? Also, how about its use for delivery of foreign genes

Line 242-43: does it really take months to recover infectious virus? If so please provide appropriate reference/s

Line 252: what do you mean unanticipated errors? do you mean to say the original reverse genetics systems had mutations in the N gene, which resulted in lower efficiency in virus recovery?

Line 255: Provide reference

Line 378: Please provide correct reference for avian borne virus. Reference the original paper that isolated the virus

Reviewer 2 Report

The manuscript reviewed the recent progresses in reverse genetics techniques of Borna disease virus 1. The word is a major contribution to this field. However, there are some issues needed to be addressed.

1. In Fig1, the reporter CAT gene was inserted in two different orientations by different researchers, the authors need to summarize the purposes  and the differences.

2. The authors have published a similar review article in 2020, please make sure there is no repetition with current manuscript.

3. As outbreaks of Borna disease virus in birds have been increased in recent years, the reviewer suggests the authors expand the related information in the part of "6.2. Reverse genetics system for other mammalian and avian bornaviruses".
